# A mixed culture of bacterial cells enables an economic DNA storage on a large scale

Min Hao[1,2,4], Hongyan Qiao 🄳 [1,2,4], Yanmin Gao 🄳 [1,2,4], Zhaoguan Wang[1,2], Xin Qiao[1,2], Xin Chen[3] & Hao Qi 🄳 [1,2✉]

DNA emerged as a novel potential material for mass data storage, offering the possibility to cheaply solve a great data storage problem. Large oligonucleotide pools demonstrated high potential of large-scale data storage in test tube, meanwhile, living cell with high fidelity in information replication. Here we show a mixed culture of bacterial cells carrying a large oligo pool that was assembled in a high-copy-number plasmid was presented as a stable material for large-scale data storage. The underlying principle was explored by deep bioinformatic analysis. Although homology assembly showed sequence context dependent bias, the large oligonucleotide pools in the mixed culture were constant over multiple successive passages. Finally, over ten thousand distinct oligos encompassing 2304 Kbps encoding 445 KB digital data, were stored in cells, the largest storage in living cells reported so far and present a previously unreported approach for bridging the gap between in vitro and in vivo systems.

[1] School of Chemical Engineering and Technology, Tianjin University, Tianjin, China. [2] Key Laboratory of Systems Bioengineering (Ministry of Education), Tianjin University, Tianjin, China. [3] Center for Applied Mathematics, Tianjin University, Tianjin, China. [4]These authors contributed equally: Min Hao, Hongyan Qiao, Yanmin Gao. ✉email: haoq@tju.edu.cn

As a biological material carrying genomic information, DNA has demonstrated great potential for long-term high-density information storage in its nucleotide sequence. The increasing capability of high throughput chip synthesis-based writing and next-generation sequencing-based reading technologies greatly advanced the development of synthetic nucleic acid mediated archival storage. Simply put, information can be synthesized into DNA oligo molecules and then read out by sequencing. To date, a number of systems have been developed for storing large archival data in synthetic oligo pools[1,2]. Classical electronic communication and computing algorithms such as Fountain and Reed-Solomon code have been adapted for conversation of digital binary information to the four-letter nucleotide sequence and error correction[3,4]. Restricted by current high-throughput oligo synthesis techniques, oligos with lengths ranging from 100 to around 200 nts are the main material for information storage in test tube. However, the oligo size fits well with the major commercial sequencing platforms, such as Illumina[5], by which sequences from 50 to 200 nucleotides can be obtained in single reads from one oligo terminus. Furthermore, chip-based DNA synthesis is the cheapest available technology, at least one or two orders of magnitude cheaper than traditional column-based oligo synthesis. Thus far, the capacity of in vitro systems with a synthetic oligo pool as the storage medium has been largely expanded, allowing up to 200 MB of information storage[4].

In contrast to test tubes, microbial cells are able to carry out the synthesis DNA material with many advanced features for archival information storage. Compared with the cell-free in vitro system, the genomic maintenance mechanism of living cells ensures that the DNA molecules are replicated with high fidelity, so that higher stability and longer storage periods could be expected. Moreover, the DNA copy rate is several orders of magnitude higher than general in vitro replication methods, such as PCR. These advanced features make living cells an attractive material for copying and distribution of information at low cost. Synthetic DNA fragments encoding archival data have successfully been inserted into the genome of various organisms, including *Escherichia coli*[6,7], *Bacillus subtilis*[8] and yeast[9]. Molecular tools were developed to engineer various DNA maintenance and genome modification systems, including reverse-transcription[10,11], recombinase[12,13] and CRISPR-Cas[14,15], for directly writing archival data into the genome in a highly controlled fashion. Moreover, circular plasmids were designed for carrying information as well, and the high copy number of some plasmids in microbial cells could facilitate the recovery of DNA material.

The in vitro and in vivo DNA storage approaches were largely developed as mutually independent system. In vitro systems rely on very large numbers of short oligos, reaching up to $10^{10}$ distinct strands[16] from microchip synthesis, These oligos are read out via a straight-forward reading workflow comprising PCR amplification and NGS sequencing[17,18]. By contrast, cells are technically able to store much larger DNA fragments, and researchers used *E. coli* cells to save DNA fragment encoding hundreds of kilobase pairs cloned from the human genome early on[19]. However, being limited by current technological capabilities, the synthesis of large DNA fragments, generally over a thousand nucleotides, is a highly time-consuming and expensive procedure[20]. Even though entire bacterial chromosomes were completely synthesized de novo[21,22], it is vary labor-intensive to carefully design the oligo units and it generally takes a long time, even several months, to build them into large fragments[23]. Moreover, it is relatively complicated to efficiently transform cells with large DNA fragments. Thus far, in vivo DNA storage has only been tested on a relatively small scale, no larger than a few thousand nucleotides[24],

far smaller than in vitro systems. Considering the storage capacity, large oligonucleotide pools have advantages in their ease of scale-up and synthesis cost. However, DNA storage inside cells has distinct advantage in terms of stable DNA maintenance for long periods of time and very low cost of replication[2].

Here, we demonstrated that a mixed culture of bacterial cells carrying large oligonucleotide pools is an economical and sustainable material for stable information storage, capable of storing DNA oligos spanning hundreds of nucleotides in length from high-throughput chip-synthesis. The BASIC code system, a DNA-mediated distributed information storage system previously developed in our lab, was applied to translate digital binary information to nucleotide base sequences with an encoding redundancy of 1.56% at software level to tolerate the physical dropout of minority oligos. By pushing the limits, oligo pools comprising 509 and 11520 distinct oligos, generating the largest reported population in a mixed culture of bacterial cells, were stored. To cover very large oligo populations, we assembled them in a redundant fashion and then stored them in a mixed culture on solid plates or in liquid medium. Furthermore, the underlying principle of the manufacturing of data storage cells was explored using specifically developed deep bioinformatic analysis tools. The results demonstrated that oligo homology assembly has relatively high bias for the sequence context, and the oligo copy number distribution became increasingly skewed as the number of fragments in the assembly increased. Nevertheless, after the assembly and transformation, we found that the large number of oligos remained stable in the mixed culture of *E. coli* cells even over multiple passages and maintained the quality of digital data for perfect information decoding. Finally, it could be demonstrated that this simple material based on a mixed culture of bacterial cells achieved in vivo storage of 445 KB of digital files in a total of 2304 Kbps of synthetic DNA in a fast and economical way. To our best knowledge, this is the largest scale archival data storage in living cells reported so far, paving the way for biological data storage taking advantage of both in vitro synthesis capacity and the biological power of living cells in an economical and efficient way, which is crucial for developing practical cold data storage on a large scale.

## Results

**DNA data storage in a mixed cell culture**. Thus far, oligo pools comprising a large set of distinct oligonucleotide sequences were used as the material for storing archival data in the major in vivo DNA storage approaches. We aimed to combine the advantage of both in vitro oligo pool mediated data storage and in vivo cell systems with a previously unreported design strategy to improve the DNA material for data storage. As illustrated in Fig. 1, binary sequences of archival data were encoded into nucleotide base

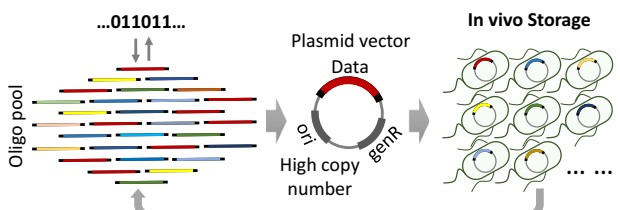

**Fig. 1 Illustration of the mixed culture of bacterial cell for large-scale data storage.** First, binary digital information was translated into nucleotide sequences using the BASIC encoding system, and then synthesized as a large short-oligo pool by chip-based high-throughput synthesis. The oligo pool was assembled into circular plasmids and then introduced into bacterial cells for stable data storage. The oligo pool could be retrieved from the mixed culture of cells for information decoding when needed.

sequences and split into groups of oligo strands a few hundreds of nucleotides in length using a BASIC code (Supplementary Note 1), which was developed for distributed DNA oligo pool information storage[25]. In this encoding system, a relatively low coding redundancy of 1.56% is enough to tolerate the physical loss, or dropout, of whole oligos. Thus, information could be perfectly decoded as long as more than 98.44% of the designed oligos can be retrieved. In addition, oligo strands with mutations including base substation or insert/deletion could be corrected by predesigned coding algorithms[25]. Following sequence encoding, the oligos were physically synthesized using the emerging high-throughput chip-based synthesis approach. Currently, there are only a few commercial products available for high-throughput oligo synthesis, and the quality of oligo pools varies with each manufacturer, and even with each batch. As reported in many previous studies, the unevenness of molecular copy number in oligo pools caused serious problems in using DNA as material for data storage[26]. For storing oligos in living cells, they should be assembled into high-copy-number plasmid vectors using homology-based cloning, without any specific sequence, and then the large population of plasmids could be transferred into an established *E. coli* engineering strain to be stored as a mixed culture. Thus, the oligo pool could simply be converted to a living cell-based material for data storage.

Mixed culture is an established approach in metabolic engineering and directed evolution, where it is used to generate DNA libraries with large diversity in living cells. However, data storage requires cells to stably carry these digital DNA sequence in large numbers without loss, and there is still a lack of systematic analyses of the stability of mixed cultures carrying large numbers of oligos. Therefore, a multistep process, including homology assembly, transformation and mixed culture, was designed to construct the living cell-based DNA storage library. For increasing the homology cloning efficiency, the homology arms were designed to reduce secondary structures and cross recognition using NUPACK (Supplementary Figs. 1 and 2)[27]. The homology arm was fused with the oligo by PCR amplification through the uniform adapter on both sides (Fig. 2a). In the oligo unit (Supplementary Fig. 3a), the main sequence of 128 nucleotides in a pool of 11520 encoded the digital information, while the corresponding address information and RS code were respectively written in 16 and 8 nts sequences located on each side of the information-carrying sequence. For identifying the direction of each sequenced oligo, "A" and "AA" tags were inserted in fixed positions in the address and RS code sequence. The sequences including the address, digital information and RS code were denoted together as payload. The adapters I and II, as well as the primer for PCR amplification, were designed outside of the two termini. The sequence carried by the payload was designed using the BASIC code algorithm with specific criteria (Supplementary Fig. 3b), including a GC content between 35 and 65%, as well as avoiding homopolymer sequences, the *Not* I endonuclease recognition sequence "GCGGCCGC" and the last 6nts of amplification primers, to minimize false priming in PCR. The designed sequence was further analyzed to avoid putative bacterial promoters using general promoter prediction algorithms[28]. In the amplified structure, two *Not* I cleavage sites were designed on both ends, by which the original oligo sequence could be directly cleaved out from the vector. A redundant assembly is designed to increase the foreign DNA load on each vector. A total of 6 homology arm sequences were designed for homology-based assembly of multiple fragments into a single vector plasmid (Supplementary Fig. 4). Oligos fused with different combinations of homology arms could be assembled together. Therefore, in a single vector plasmid, 1F, 3F and 5F of fragments could be assembled and each fragment could cover the intact oligo pool. Thus, the multifragment assembly principally could greatly increase the chance of oligos being assembled into the vector plasmid. Following the assembly, circular DNA will be introduced into *E. coli* DH10β cells for mixed culture, and then the large number oligos could be retrieved from isolated plasmid DNA.

**Mixed culture of cells carrying a redundant assembled large oligonucleotide pool.** Firstly, we tested a pool comprising 509 distinct oligos as part of a chip-synthesized pool. It is known that cells lose some sequences from the population due to disadvantages of some clones in the growth rate in mixed culture[29,30]. In view of the possible loss of cells carrying minority oligos in the pool, electrically transformed cells were cultured on the surface of solid medium, which should give all cells carrying the assembled plasmids equal chance to form colonies. The colony number assembled from 1F assembly of a total of 0.08 pmol oligo fragments and 0.16 pmol vector was almost twice that of the 3F (assembly of 0.8 pmol oligo fragments and 0.16 pmol vector) and 5F (assembly of 0.8 pmol oligo fragments and 0.16 pmol vector) on the solid medium surface (Fig. 2b and Supplementary Figs. 5–7). There is a trade-off between the assembly efficiency and capacity, and redundant assembly could increase the loading capability for each vector, but largely decrease the assembly efficiency. Totally, 122.4 and 158.6 and 268 copies per designed oligo was calculated from the counted colony number for 1F, 3F and 5F respectively. After plasmid isolation, the oligo pool was directedly cut out using the site-specific endonuclease *Not* I (Supplementary Figs. 8–10) and sequenced by standard NGS. Errors including substitutions or indels were counted, and it was observed that the frequency of substitutions was higher than that of indels for all of the assembly samples (Fig. 2c, Supplementary Note 6). Importantly, the error rate was consistent with previous studies. It was also observed that sequencing reads with single letter errors (substitutions or indels) were much more frequent than other types of error (Supplementary Fig. 11, Supplementary Note 6), which is in agreement with our previous study as well[4]. For all of the assembly samples, the oligo was 100% identified in the sequencing reads, but 1F assembly recorded the lowest minimal necessary coverage of sequencing reads, at which perfect 100% oligos can be identified (Fig. 2d and Supplementary Fig. 12). After the success of oligo retrieval using solid culture, mixed culture in liquid medium was also tested (Supplementary Fig. 13). Plasmid DNA was isolated from 5 ml of liquid mixed cell culture and sequenced. The minimal necessary coverage was even lower than that of the 1F assembly on solid surface (Fig. 2d). Furthermore, the frequency for each oligo in the retrieved pool was quantified and a similar frequency distribution was observed for all the assembly samples with very close Gini indices (Supplementary Figs. 14 and 15). These results demonstrated that the DNA pool of 509 distinct oligos was stably stored in the mixed culture.

Next, a DNA pool comprising 11520 distinct oligos with a length of 200 nucleotides, over 20 times larger than the first pool, was tested. About 445 KB of digital files were encoded, including images, word text and various types of files (Supplementary Fig. 3c). It was observed that the mixed culture in liquid medium gave a lower minimal necessary coverage of sequencing reads than the solid culture. Additionally, subculture is necessary for long-term storage at low cost. Therefore, the DNA pool with 11520 oligos was assembled to test the subculture of this very large cell population (Fig. 3a). In total, the mixed culture was successively passaged 5 times, and plasmids carrying the digital information were isolated from a large liquid culture and then a large number of oligos was recovered following *Not* I digestion

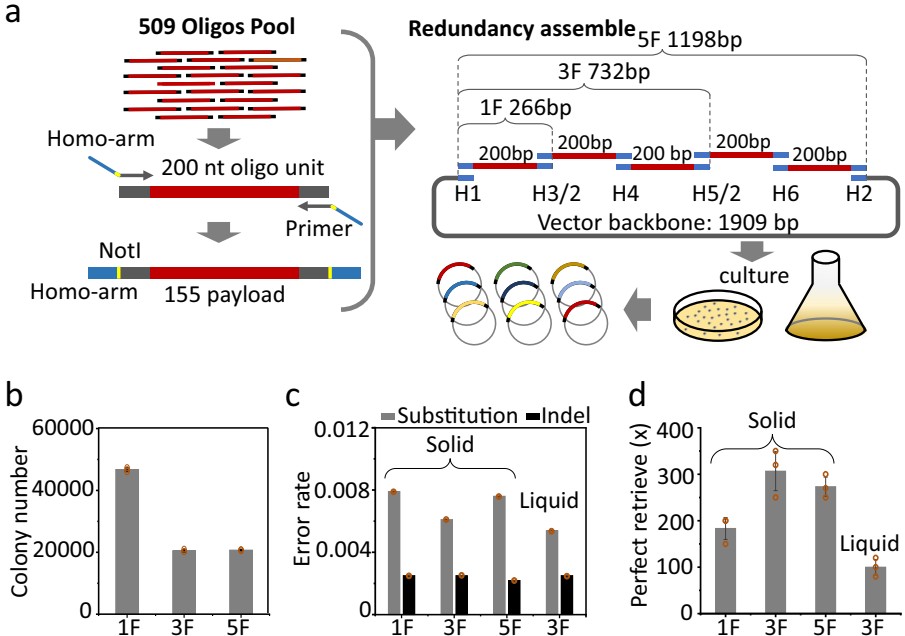

**Fig. 2 Redundant assembly of 509 oligos pool for storage in mixed cell culture. a** Schematic of the workflow for the assembly of a DNA pool comprising 509 distinct oligos. Oligos were fused with homology arms via PCR amplification, *Not* I cleavage sites were added for oligo retrieval afterwards. Multiple insert fragments, 1F insert fragment, 3F for three insert fragments and 5F for five insert fragments, respectively, each fragment potentially containing any of the 509 oligos, were assembled into a vector plasmid backbone of 1909 bps using off-the-shelf homology assembly reagents. Finally, the assembled plasmids were introduced into *E. coli* cells for mixed culture on solid plates or in liquid medium. **b** The colony number was counted on the solid medium surface for 1F, 3F and 5F assembly. **c** Letter error, base substation or indel (both base insertion or deletion) occurred in the oligo pool retrieved from mixed culture on solid plates or in liquid medium and was quantified as percentage of counted error base number vs. the total sequenced base; substation errors in gray bars, indel errors in dark bar. **d** The minimal necessary sequencing reads depth for perfect retrieval of all 509 oligos from 1F, 3F, and 5F assembly samples on solid plates or in liquid medium. Error bars represent the SD, where $n = 3$.

(Supplementary Fig. 16). There was no obvious difference in the error rate even between the 1st and 5th subculture of 1F or 3F assembly samples (Table 1). In agreement with previous results, the frequency of substitutions was still higher than that of indels. Compared with the original master pool, the error rate was in the same order of magnitude. It demonstrated the high fidelity of DNA amplification inside living cells. Among the NGS sequencing reads, some sequences were identified as contamination from the host cell genome by deep bioinformatic sequence analysis, but the contamination content was very low, less than 0.2% of the total sequencing reads. This contamination may come from the plasmid isolation step, because there are also 20 *Not* I cleavage sites on the DH10β genome. However, it is very easy to distinguish these contaminations from the true digital oligo sequence based on the designed adapter sequence on the oligo terminus (Supplementary Fig. 3a). Because the digital DNA sequence was stored on a plasmid, it is still relatively easy to remove the host cell genome contamination during the isolation process using available commercial bio-reagents. This could be another advantage compared to approaches in which digital information is stored directly on the bacterial chromosome.

Interestingly, the population of assembled plasmids carrying the inserted DNA sequences encoding digital information remained relatively stable. The frequency of each oligo in the pool did not change largely between the 1st and 5th passages of the 1F or 3F assembly samples (Fig. 3b and Supplementary Fig. 17, Supplementary Note 4). Moreover, the dropout rate decreased with deep sequencing (Fig. 3c and Supplementary Fig. 18). The bioinformatic analysis therefore demonstrated the stability of the oligo pool recovered following successive passaging. Remarkably, the mixed culture of *E. coli* cells carrying this large population of oligos remained content, with a Gini

coefficient of 0.41 and 0.48 in the 1st and 5th passages of the 1F assembly sample, respectively (Supplementary Table 2). By contrast, the content uniformity was skewed largely in the 3F assembly sample (Fig. 3d). Compared with the 1F assembly, about 21% of oligos in the 3F assembly were enriched, accounting for up to 96.2% of the total sequencing reads, while the remaining 79% of oligos were largely depleted, accounting for only 3.8% of the sequencing reads, corresponding to a very high Gini coefficient of 0.87 (Fig. 3e). These results reflected the influence of sequence complexity on the random assembly process, in which 3F assembly was more vulnerable due to the high sequence complexity of the large oligo pool. Compared with the 1F assembly, the number of DNA assembly events was double in 3F assembly and resulted in more bias in the finial assembled DNA population. However, the 1st and 5th passages of the 3F assembly sample were relatively consistent with similar Gini coefficients and oligo frequencies. The stable oligo frequency distribution, even across multiple passages, indicated that the mixed culture of living cell could be used as a material for data storage.

**Large-scale DNA data storage in living cells.** We succeeded in establishing a method for DNA data storage in living cells via a simple multiple-step process, by which a DNA pool comprising a large number of oligos could be quickly transferred into living cells for data storage (Fig. 4a). Furthermore, deep bioinformatic analysis explored the underlying principles of this digital storage cell manufacturing process. The assembly was revealed to be a biased process, its efficiency going down with increasing assembly fragment number in the designed redundant sequence set. For the 11520 DNA sequence pool, largely fewer colonies were counted in the 3F than in the 1F assembly sample, and average copy number

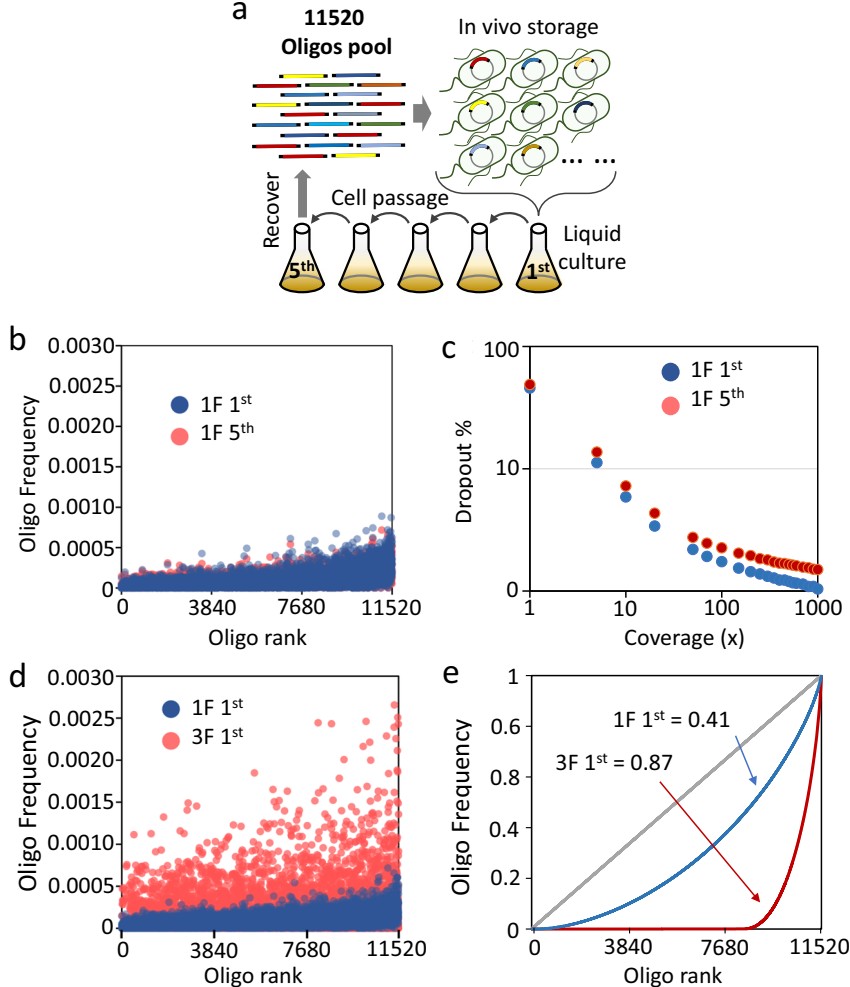

**Fig. 3 Mixed culture of cells carrying a redundant assembly of 11,520 oligos for large-scale data storage. a** Schematic of cells carrying an assembled pool of 11520 oligos for successive multiple subculture. Cells collected from 1st and 5th passage were subjected to oligo retrieval and information decoding. **b** The frequency for each of the 11520 oligos quantified in sequencing reads from the 1st (blue dots) and 5th (red dots) passage of the one fragment (1F) assembly sample. **c** The oligo dropout rate was quantified at different sequencing depths (various amounts of NGS sequencing reads) of the 1st (blue dots) and 5th (red dots) passage of the one fragment (1F) assembly sample. **d** The frequency for each of 11,520 oligos quantified in sequencing reads from the first culture passage of the one insert fragment assembly (1F, blue dots) and three insert fragment assembly (3F, red dots). **e** The Gini coefficient was calculated for the oligo frequency distribution in the retrieved oligo pool. The 1st passage of one fragment assembly was calculated to have a Gini of 0.41 (blue line) and the 1st passage of the three fragments assembly (red line) had a Gini of 0.87.

**Table 1 The letter error rate was quantified in the sequenced oligos of the 1st and 5th subculture passages of the one insert fragment (1F) or three insert fragment (3F) assembly.**

|  | 1F 1st | 1F 5th | 3F 1st | 3F 5th | Master pool |
|---|---|---|---|---|---|
| Substitution nt% | 7.6E−03 | 6.4E−03 | 4.8E−03 | 8.5E−03 | 2.1E−03 |
| Indel nt% | 3.1E−04 | 3.0E−04 | 3.2E−04 | 3.4E−04 | 4.0E−04 |
| Genomic contamination read% | 1.1E−03 | 7.9E−04 | 6.7E−04 | 1.1E−03 | 0 |

The amount of oligos with sequences of high similarity with the host cell genome among the sequencing reads was identified as genomic contamination.

per designed oligo was calculated to be 9.42 for 1F, compared to only 0.91 for the 3F assembly sample. Thus, it took a longer time for 1st of 3F assembly cells (11 h) to reach an $OD_{600}$ of 1.2 than that of the first 1F assembly cell (8.4 h). Although over $10^6$ average molecule copies for each fragment were subjected to the assembly process, the successfully assembled copy number for each oligo was found to be only dozens to hundreds after the assembly and transformation steps. However, the mixed culture amplified the population in a relatively stable fashion without skewing the oligo frequency distribution, so that an average copy number of over $10^7$ for each oligo could be recovered from a batch culture. For the 1F subculture sample, it was possible to retrieve enough oligos (about $10^3$ copies of each oligo) for perfect information decoding, with finial 0.9 and 1.4% respective dropout rates for the 1st and 5th passages, which were both lower than the decoding limit of 1.56%. However, much more oligos were lost from the 3F assembly sample, with 26.5 and 32.8% respective dropout rates for the 1st and 5th passage. A similar retrieval rate

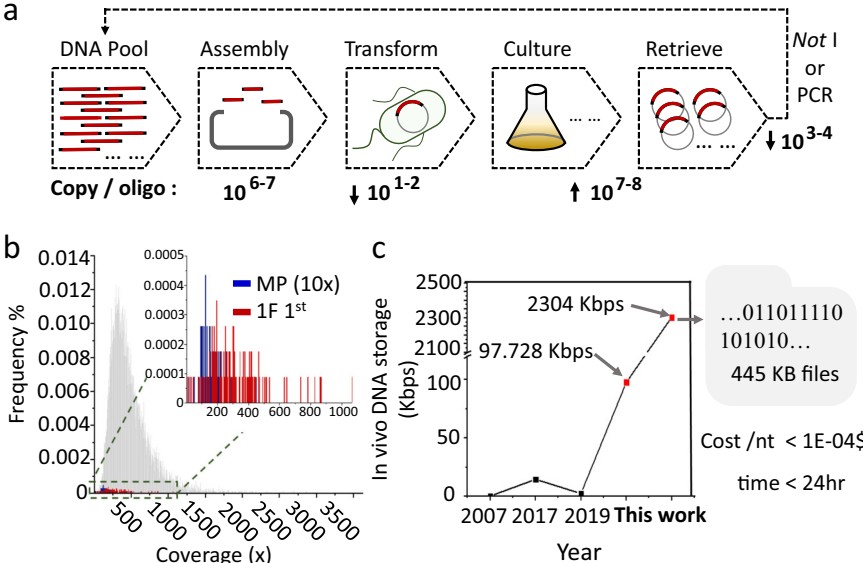

**Fig. 4 Large-scale DNA data storage in living cells. a** The workflow for the manufacture of a mixed culture living cell data storage material. The assembled oligo pool with $10^6$ to $10^7$ average copies for each oligo was subjected to assembly and then introduced into *E. coli* cell. A $10^1$ to $10^2$ average colony number for each oligo was obtained and then the cell population could be amplified to large scale in mixed culture for further plasmid retrieval and information decoding. **b** The 0.9% lost oligos in the 1st passage of the one-fragment assembly (red line) and the 0.56% lost oligos in the 10× deep sequencing reads of the original master pool (blue line) were mapped to the oligo frequency distribution of the original master pool (gray line). **c** In contrast with previous reported major systems for DNA storage in living cells, including 0.25 kbps by Yachie in 2007, 14.56 bps by Shipman in 2017 and 2.448 kbps by Sun in 2019, the total of 97.728 kbps of DNA for the 509 oligos pool and 2304 Kbps for the 11520 oligos pool stored in a mixed culture of *E. coli* cells at a cost lower than 0.001\$ per base, and the mixed cell storage material could be manufactured within 24 h.

was obtained for the oligo pool recovered by PCR amplification (Supplementary Table 3). By mapping the lost oligos from the 1F assembly into the frequency distribution of the original master pool from chip synthesis, it was found that the lost oligos from the master pool in sequencing coverage of 10× did not overlap with that of 1F, and many oligos lost from 1F were mapped to a high frequency in the master pool (Fig. 4b). Furthermore, the enriched oligos in the 1st passage of 3F were also mapped to the frequency distribution of the master pool, and this group of oligos covered a very wide area, mapping to oligos with both high and low coverage (Supplementary Fig. 19). In the 10-mer DNA sequence pattern analysis, the top 10% high frequency 10-mer pattern accounted for 42.1% of the total 10-mer pattern counts in the 1st passage of the 3F assembly sample, compared to 26.5% for the 1st passage of the 1F assembly sample, resulting in a decrease of 16.4% (Supplementary Fig. 20). The 10-mer frequency distribution was obviously different between the enriched oligo sequences (Supplementary Fig. 21). These results also supported the idea that the assembly is a biased process dependent on the sequence context rather than the oligo concentration in the original master pool. However, after the living cell material was manufactured, the mixed culture preserved the stability of digital DNA for large-scale living cell data storage.

## Discussion

DNA is considered to have high potential as an advanced material for mass data storage, a serious problem human society will face in the very near future. In addition to the storage density, the crucial features including storage longevity and low copy cost are highly dependent on biological systems inside living cells. Thus far, the data storage capacity has been demonstrated mainly using large oligonucleotide pools, up to 13 million DNA oligos produced via advanced chip-based synthesis[4]. Although several molecular tools, such as CRISPR and special recombinases, have been adapted to write information into the genome, the capacity

is still very far away from the in vitro systems, not larger than 20 kbps to date[31]. Theoretically, one intact single DNA fragment is the desirable material for data storage, mimicking the way genomes store information in nature, but the current DNA writing technology is not designed for the synthesis of long DNA strands. Although entire bacterial genomes have been reconstructed from chemically synthesized oligos[32], the synthesis of large DNA fragments is extremely time- and labor-intensive. The cost for a DNA fragment over 10 kbps is about 0.2\$/nt at the major commercial companies[33,34], and it generally takes several months to build with high risk of failure for complicated sequences. Considering the scale of application, it is very challenging for large DNA fragments to match for practical data storage needs until suitable synthesis technology is developed. By contrast, pools of oligos several hundreds of nucleotides in length can be synthesized at a cost lower than 0.001\$/nt[33], several orders of magnitudes lower than that of large-fragment DNA synthesis. Additionally, over a million distinct strands can be manufactured at same time in just a couple of business days, while the cost keeps going down with the scale-up of synthesis. Therefore, a mixed culture of bacterial cells carrying a large oligonucleotides pool could be a powerful DNA storage material with advantages of both oligo pools and living cell for data storage. In comparison with the major previous reported living cell DNA storage systems[9,31,35], the total 2304 Kbps DNA storage achieved in this study is the largest storage size of data, including text, image documents and computer program code, in living cells (Fig. 4c and Supplementary Note 7). In comparison with storing long DNA fragments on the chromosome, mixed culture storage materials could be fabricated within 24 h after oligo pool synthesis at a total manufacturing cost, lower than 0.001\$ per base (Supplementary Note 3). Additionally, a large DNA fragment imposes a greater burden on cell growth. In mixed cell culture, each cell only carries one small fragment of informational DNA, whose sequence was also analyzed by bioinformatic tools to avoid

potential bacterial promotor sequences[28]. The stable passaging also indicted that these artificial digital sequences did not interference with the host cell much, and oligos carrying toxic sequences will not interfere with others, which is another benefit of this mixed culture strategy. Thus, in view of this very artificial purpose, digital information storage, it is not necessary to follow the modalities by which genomic information is recorded in nature.

Mixed culture is a strategy that has been successfully applied in many fields. In metabolic engineering, different types of microbial cells were cultured together for mutual metabolic benefit[36,37], but the size of the population is relatively small. Larger DNA libraries encoding enormous genomic diversity were generated in living cells for screening of specific biological functions in directed evolution research[38,39]. Although a large DNA library has been created in living cells to generate very large phenotypic diversity, stably carrying these large numbers of DNA structures is not necessary. Generally, it is difficult to balance the growth rate between different cells. In the present work, even in one insert fragment assembly of the large oligonucleotide pool, there is at least 11520 genotypes, and there will be a huge number in the redundant assembly of multiple fragments, the largest mixed culture reported so far. As demonstrated by statistical analysis, the assembly process is biased due to the high sequence complexity in the large oligo pool, but it could be improved by more specific homology arm design, and more effective in vitro DNA homology assembly methods. Furthermore, the thermal process of DNA origami assembly could be recruited to improve the assembly efficiency. However, a relatively stable mixed culture was achieved even after multiple passages. The copy number distribution of oligos remained stable with very similar value of the Gini coefficient in the successive multiple passages of the mixed culture (Fig. 3c and Supplementary Figs. 15, 22–23, Supplementary Table 2). There are a few possible reasons for this stability. The artificial purpose of storing digital information allows the design of sequences that avoid sensitive sequence patterns with specific biofunctions, e.g., polynucleotides (polyA etc.) and specific endonuclease recognition sequences (Supplementary Note 2). The bioinformatic analysis demonstrated that there is no sequence similarity between the designed oligos and the whole E. coli DH10β genome, with e-values of $10^{-6}$ (Supplementary Note 5). It was demonstrated that the digital DNA sequence has no large influence on either host cell growth or vector plasmid replication. Additionally, storing digital sequences on plasmids decreased the information contamination from the genome. Therefore, this simple method is highly compatible with any oligo pool for data storage, and scale-up could be achieved easily in a parallel manner based on the greater than $10^4$ oligo storage capacity we demonstrated here.

When manufacturing living cell material for data storage, assembly and transformation become crucial steps in determining the actual size of the oligo population. Deep bioinformatic analysis demonstrated that the assembly process is biased based on the sequence context and transformation is a relatively random and inefficient process. Consequently, the size of the oligo population decreased almost two orders of magnitude. The bias introduced during assembly and transformation is highly dependent on the used bioreagent, and homology assembly methods should be re-designed to improve their efficiency for assembly of oligo pools with large molecular populations. In addition, it was found that the dropout rate during mixed culture showed a good correlation with the dropout curve of the master oligo pool, which could be quantified to assess the manufacture of storage material (Supplementary Fig. 24). Therefore, there is still much space to improve the capability of mixed cell cultures for storing data. The unevenness of oligo copy number in the original

chip-synthesized DNA pool was obvious, which is also a serious problem for in vitro DNA storage approaches[26]. Oligos with high copy number have a higher chance of being integrated during the assembly process. As demonstrated, the oligo pool retrieved from cultured cells is highly unbalanced. Revising the oligo copy number of the oligo pool close to make it more even could improve the biased assembly process. The unevenness in oligo copy number caused oligo dropout in both the assembly and cell culture process. Recently, an approach has been developed to balance the oligo mix using a specifically designed primer mix[40]. However, it still needs more development before it can be applied to DNA pools for storage. Therefore, more synthetic tools could be developed to improve the chip-synthesized oligo pool and foreign DNA transformation, and balance the large mixed culture. In summary, a DNA oligo pool from chip synthesis comprising over ten thousand strands was quickly transferred into living cells for data storage. The resulting mixed culture of E. coli cells is a stable material for storing a large number of DNA sequences encoding digital data. To our best knowledge, here we achieved the largest data storage in living cells to date.

## Methods

**Strains and culture conditions**. Electrocompetent competent E. coli DH10β was used for cloning and purchased from Biomed Co., Ltd. (Beijing, China). The antibiotics of ampicillin were used at 100 mg/mL. Unless otherwise noted, the cells were cultured in Luria-Bertani (LB) broth with shaking at 220 r.p.m. or on LB agar plates at 37 °C.

**Library construction**. For the assembly of 509 sequences, the oligo pool was synthesized and the lyophilized pool consisted of 11,776 oligos of 192 nts (synthesized by Twist Bioscience), which included the 152 nts payload in each oligo. The pool was resuspended in 1× TE buffer for a final concentration of 2 ng/μL. One of the files, 509 oligos, was flanked by binding sites for the premixed primers F02/R02. PCR was performed using Q5® High-Fidelity DNA Polymerases (NEB #M0491) and primers F01-F04/R01-F04 (10 ng oligos, 2.5 μL of each primer mix (100 mM), 0.5 μL Q5 High-Fidelity DNA Polymerase, 4 μL 2.5 mM dNTPs in a 50 μL reaction). Thermocycling conditions were as follows: 5 min at 98 °C; 10 cycles of: 10 s at 98 °C, 30 s at 56 °C, 30 s at 72 °C, followed by a 5 min extension at 72 °C. The library was then purified using Plus DNA Clean/Extraction Kit (GMbiolab Co, Ltd. #DP034P) and eluted in 40 μL ddH2O. This library was considered the master pool and run on a 2% agarose gel to verify the correct size. For the assembly of 11520 sequences, the synthetic DNA pool consisted of 11520 oligos of 200 nts (synthesized by Twist Bioscience), which included the 155 nts payload flanked by binding sites for the primers F1/R1 (Supplementary Fig. 3). The lyophilized pool was rehydrated in 1× TE buffer and the above protocol was used to amplify the file.

**Computational strategy for primer homologous arm design**. The primers were included three parts, one was the homologous arm (Homo-arm) which used for gibson assembly, the other was Not I which endonuclease recognition site and the last was address sequence. The primer design algorithm was implemented to codify rules for assembly fragments production using PCR. Primer homologous arm was designed using the NUPACK (http://www.nupack.org). Sets of primers are identified satisfying the following rules for homologous arm design: (1) homologous arm length 25 nt; (2) guanine-cytosine (GC) content between 40 and 60%; (3) no interaction between each homologous arm. Selected primers passing all of the aforementioned tests are provided as output as correct pairs that would yield the product of the defined size (Supplementary Figs. 1 and 2).

**Assembly experiment**. For the backbone preparation, the pUC19 plasmid was used as templates for PCR. The PCR was performed using PrimeSTAR® Max DNA Polymerase (Takara #R045Q) and primer set PCR-vactor-F/R. Thermocycling were amplified for 30 cycles, involving denaturation for 15 s at 98 °C, annealing for 5 s at 55 °C and primer extension for 20 s at 72 °C. Then the PCR products were purified by gel cut using Plus DNA Clean/Extraction Kit (GMbiolab Co, Ltd. #DP034P) and eluted in 30 μL ddH2O.

For the 509 oligos pool assembly fragment preparation, we started with the master pool as described above. The fragments were prepared with different homologous arms using Q5® High-Fidelity DNA Polymerases and the corresponding primers (the sequence was showed in Supplementary Table 1). The template used 0.2 ng of oligo input from the master pool into 50 μL PCR reactions. Initial denaturation was carried out at 98 °C for 5 min. Follow this by 20 PCR cycles, involving denaturation for 30 s at 98 °C, annealing for 30 s at 56 °C and primer extension for 20 s at 72 °C. Finally, the PCR reaction was terminated by

incubating the solution at 72 °C for 5 min. The library was then purified using Plus DNA Clean/Extraction Kit (Gmbiolab, #DP034P) and eluted in 30 μL ddH$_2$O. The final library was run under the same conditions as described. Then the Gibson Assembly® Master Mix—Assembly (NEB, #E2611) was used according to user's manual.

For the 11520 oligos pool assembly fragment preparation, we started with the master pool as described above. The fragments were prepared with different homologous arms using 2× EasyTaq® PCR SuperMix (AS111, TRANS) and the corresponding primers (the sequence was showed in Supplementary Table 1 and Supplementary Fig. 4). The template used 20 ng of oligo input from the master pool into 50 μL PCR reactions. Initial denaturation was carried out at 94 °C for 2 min. Follow this by 10 PCR cycles, involving denaturation for 30 s at 94 °C, annealing for 30 s at 53 °C and primer extension for 20 s at 72 °C. Finally, the PCR reaction was terminated by incubating the solution at 72 °C for 5 min. The library was then purified and eluted in 100 μL ddH$_2$O. NEBuilder® HiFi DNA Assembly Cloning Kit (NEB, #E5520) was used according to user's manual.

The concentration of each fragment was calculated for optimal assembly, based on fragment length and weight, we used the following equation:

$$pmols = (weight\ in\ ng) \times 1000/(base\ pairs \times 650\ daltons). \quad (1)$$

Based on this calculation, the number of molecules per oligo and copy number was determined according to the following formula:

$$moles\ dsDNA\ (mol) = mass\ of\ dsDNA\ (g)/((length\ of\ dsDNA\ (bp) \times 607.4) + 157.9\ g/mol; \quad (2)$$

$$DNA\ copy\ number = moles\ of\ dsDNA \times 6.022e23\ molecules/mol. \quad (3)$$

We found an excellent agreement concentration between the assembly fragment and the backbone of Gibson assembly experiment. Thus, for 509 oligos pool assembly experiment, optimized cloning was performed using $10^{11}$ copy number of vectors and $10^8$ copy number of inserts. For 11,520 oligos pool assembly experiment, optimized cloning was performed using $10^{10}$ copy number of vectors and $10^{10}$ copy number of inserts. The sample was incubated in a thermocycler at 50 °C for 60 min, respectively. Following incubation, store samples on ice or at −20 °C for subsequent transformation.

**DNA storage in living cells**. To prepare the fragments for the 509 oligos pool assembly, we started with the master pool as described above. The fragments were prepared with different homologous arms using Q5® High-Fidelity DNA Polymerase and the corresponding primers. Then, the Gibson Assembly® Master Mix (NEB, #E2611) was used according to the manufacturer's instructions. To prepare the fragments for the 11520 oligos pool assembly, we started with the master pool as described above. The fragments were prepared with different homologous arms using 2× EasyTaq® PCR SuperMix (AS111, TRANS) and the corresponding primers. NEBuilder® HiFi DNA Assembly Cloning Kit (NEB, #E5520) was used according to the manufacturer's instructions.

**Transformation and culture**. Electroporation was carried out in 1-mm gap cuvettes with conditions 1.8 kV, 200 Ω, 25 Mf, cells were recovered into fresh SOB medium for 1 h at 37 °C. For 509 assembly experiment, each sample (5 μL) was added to DH10β electrocompetent cells (50 μL) for electroporation reaction.

After recovery, 500 μL cells were plated on the selection medium plates with appropriate selective condition (Amp) and the colonies were counted by ImageJ. For 11520 assembly experiment, each sample (2 μL, totally 20 μL) was added to DH10β electrocompetent cells (50 μL) for electroporation reaction. After recovery, 500 μL cells were plated on the selection medium plates with appropriate selective condition (Amp) and the colonies were counted by ImageJ. The transformation rate was calculated by the following equation:

$$Transformation\ efficiency (cfu/\mu g) = colonies\ on\ plate/plasmid\ DNA\ spread\ on\ plate \quad (4)$$

$$Transformation\ rate = Transformation\ efficiency/10^{10} (cfu/\mu g)^* \quad (5)$$

*note: $10^{10}$ cfu/μg is the theoretical transformation efficiency of DH10β electrotransducer cells.

For 509 oligos pool assembly experiment, another 500 μL cells were inoculated in 5 mL Luria broth (LB) medium plus appropriate antibiotics and grown overnight [37 °C, 220 revolutions per minute (RPM)] to obtain seed cultures. For 11520 oligos pool assembly experiment, 5 mL recovered cells were inoculated in 45 mL Luria broth (LB) medium plus appropriate antibiotics and grown overnight to obtain seed cultures. The seed cultures were then serially diluted (1:10) in 50 mL of prewarmed LB plus appropriate antibiotics followed by OD$_{600}$ reached 1.2. This consecutive procedure was repeated 5 times (Supplementary Fig. 12).

**Data recovery**. After liquid and plate culture, the plasmid library was extracted using a plasmid miniprep Kit (TIANGEN, #DP103). Then, QuickCut™ Not I (Takara, #1623) was used for fragments recovery. After gel recovery of the correct fragments using the Plus DNA Clean/Extraction Kit, the samples of the 509 oligos pool (1F, 3F and 5F) and the 11520 oligos pool (passage-1 and passage-5 of 1F and

3F) were sequenced directly. To obtain more complete information, we performed a PCR of the constructed plasmid to amplify the 11,520 oligos pool (passage-1 and passage-5 of 1F and 3F) using Q5® High-Fidelity DNA Polymerases and the primer pair F02/R02. The thermocycling protocol was: (1) 98 °C for 5 min, (2) 98 °C for 30 s, (3) 54 °C for 30 s, (4) 72 °C for 10 s, then repeat steps 2–4 five times, followed by a final elongation at 72 °C for 5 min. The products were purified using the Plus DNA Clean/Extraction Kit (GMbiolab Co, Ltd. #DP034P) and sequenced.

**Statistics and reproducibility**. Experiments with various sizes of DNA oligo pools were performed with at least 3 separated samples. Histograms were generated using Origin software. Quantitative data in figures are presented as the mean and standard deviation from three biological replicates.

**Reporting Summary**. Further information on research design is available in the Nature Research Reporting Summary linked to this article.

## Data availability
The data supporting the conclusions of this article are included within the article and its additional files. Furthermore, the original sequencing FASTQ file and the designed sequence file may be obtained via (http://pan.tju.edu.cn:80/#/link/FBECCB92999B055C2C2618A1B386F192, Code: 9ztM and http://pan.tju.edu.cn:80/#/link/3739C996782A37FB98440298DEBF29DA, Code: YQ0a). Source data underlying plots shown in figures are provided in Supplementary Data 1.

## Code availability
The BASIC code for encoding and decoding for both Linux and Windows and bioinformatic analysis programs may be obtained from this link (https://biorxiv.org/cgi/content/short/2020.02.09.940411v1).

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

## Acknowledgements

This work was supported by the National Science Foundation of China (Grant Nos. 21476167, 21778039 and partly by 21621004) and the National Key R & D Program of China (2019YFA0904100).

## Author contributions

M.H., Y.G. and H.Qiao. contributed equally to this work. M.H. was involved in a part of the experiments and supplementary information writing. H.Qiao, Y.G. did the main experiments, data analysis and contributed largely to the manuscript writing. Z.W. and X.Q. did a part of the experiments. X.C. provided a part of the bioinformatics statistical analysis programs. H.Qi designed experiments, analyzed data, contributed to manuscript writing, and supervised this work.

## Competing interests

Hao Qi is the inventor of a patent application for the biochemical method described in this article. The initial filing was assigned the Chinese patent application number 201911121023.7. The remaining authors declare no competing interests.
