## [Peer Review File · Communications Biology]

Reviewers' comments:

Reviewer #1 (Remarks to the Author):

Overview: The authors describe a method for using DNA oligonucleotides for information storage in a living bacterial cell. To do this they cloned synthetic oligonucleotides into a vector using various levels of DNA assembly containing 1, 3 or 5 inserts and transformed them into *E. coli*. The authors observed a decrease in oligonucleotide sequence recovery as the cloning complexity increased. This observation is in keeping with anecdotal evidence in DNA synthesis literature. The authors also report that the oligonucleotide libraries remain stable after 5 passages of the culture on media. The authors created two information encoded oligonucleotide libraries containing either 509 or 11520 distinct oligonucleotides which they cloned into a vector and transformed into *E. coli*. The largest library the authors claim is the largest oligonucleotide library ever stored in a living cell.

Comments: Overall the data and conclusions presented by the authors was well organized and clearly stated and supported by the presented materials and figures. The paper could use a good edit to fix some grammatical issues and improve readability. The figures are well done and the captions were highly descriptive and well done.

While the work described herein does appear to be a logical extension and step-up in complexity from previous in-cell memory storage work we remain skeptical of this and other in vivo methods for large scale storage of information due to the inherent limitations of constructing and transforming DNA into cells. As the authors report, assembling information bearing oligonucleotides into longer constructs which would in effect store more information per vector actually decreased the recoverability of the information from the plasmids isolated. As reported the most efficient method was to clone one oligo at a time. While this inefficiency could be due to the assembly method used and could likely be improved with some optimization the ultimate ceiling for in vivo methods will be the transformation efficiency and the efforts required to recover the information at a given scale compared to in vitro methods. This being said the author's work does stand as a way point for potential improvements of in vivo DNA information storage and would be of interest to the community.

To improve this work besides a mild edit for grammar and readability the authors should include an improved discussion of the oligonucleotide encoding scheme and the oligonucleotide information payload design. For example was there sequences that were excluded due to bio-incompatibility issues? Structure? Homopolymers? Cryptic RBSs, etc... Also a figure or table showing the oligonucleotide errors of the oligonucleotide pool prior to cloning and passaging through the cells would be useful to include so that the error modes can be better understood.

Nitpicks: Please don't use the phrase 'massive oligos' it is confusing. Please replace with 'large oligonucleotide pools' or something similar.

Reviewer #2 (Remarks to the Author):

This manuscript by Min Hao, et al. reported information storage in live bacterial DNA. The bacterial cells act as the information carrier and amplification media before the extra of the information. The overall concept is interesting, and the data in the manuscript proves the preliminary feasibility of this approach. Understandably, the current study has its limitations and future optimization and improvement are necessary to make this in vivo information storage more robust. For example, there are noticeable information loss and error after multiple rounds of bacterial growth, which could limit the practicality of the amplification via bacterial growth. Nonetheless, despite the limitations, I believe

this is a solid manuscript for the journal, and would recommend only a few minor changes.

1) The strategy for random assembly of large DNA pool proposed in this manuscript is interesting. As known, the assembly efficiency highly dependent on the specificity in homology arm sequence, and this proposed random assembly is designed for information storage, in which large sequence complexity will be expected. As observed in experiment of figure 3, the author has defined the assembly as biased process, so there should more discussion about how the sequence complexity contribute to bias in the assembly process and how to deal with it for stable assembly.

2) The DNA oligo material for assembly in this manuscript was synthesized from chip based massive synthesis, however, the molecule copy for each oligo from this high throughput synthesis is not uniform which has been demonstrated by many previous studies, so the unbalanced copy number should influence the random assembly of oligo pool as well. So, the author should explain more how the unbalanced copy number of DNA oligo impact on this assembly and information storage.

3) As presented in figure 3, it is very interesting to find out that the DNA oligo population was stably passaged for multiple generations, but the result also showed that oligo got lost during the passaging, although the DNA losing between the first and fifth generation is surprisingly small, I think author should give more detail analysis and discussion about how the DNA losing during the passaging.

Referee expertise:

Referee #1: Synthetic DNA

Referee #2: Information storage in DNA

Reviewers' comments:

Reviewer #1 (Remarks to the Author):

Overview: The authors describe a method for using DNA oligonucleotides for information storage in a living bacterial cell. To do this they cloned synthetic oligonucleotides into a vector using various levels of DNA assembly containing 1, 3 or 5 inserts and transformed them into *E. coli*. The authors observed a decrease in oligonucleotide sequence recovery as the cloning complexity increased. This observation is in keeping with anecdotal evidence in DNA synthesis literature. The authors also report that the oligonucleotide libraries remain stable after 5 passages of the culture on media. The authors created two information encoded oligonucleotide libraries containing either 509 or 11520 distinct oligonucleotides which they cloned into a vector and transformed into *E. coli*. The largest library the authors claim is the largest oligonucleotide library ever stored in a living cell.

Comments: Overall the data and conclusions presented by the authors was well organized and clearly stated and supported by the presented materials and figures. The paper could use a good edit to fix some grammatical issues and improve readability. The figures are well done and the captions were highly descriptive and well done.

While the work described herein does appear to be a logical extension and step-up in complexity from previous in-cell memory storage work we remain skeptical of this and other in vivo methods for large scale storage of information due to the inherent limitations of constructing and transforming DNA into cells. As the authors report, assembling information bearing oligonucleotides into longer constructs which would in effect store more information per vector actually decreased the recoverability of the information from the plasmids isolated. As reported the most efficient method was to clone one oligo at a time. While this inefficiency could be due to the assembly method used and could likely be improved with some optimization the ultimate ceiling for in vivo methods will be the transformation efficiency and the efforts required to recover the information at a given scale compared to in vitro methods. This being said the author's work does stand as a way point for potential improvements of in vivo DNA information storage and would be of interest to the

community.

To improve this work besides a mild edit for grammar and readability the authors should include an improved discussion of the oligonucleotide encoding scheme and the oligonucleotide information payload design. For example were there sequences that were excluded due to bio-incompatibility issues? Structure? Homopolymers? Cryptic RBSs, etc... Also a figure or table showing the oligonucleotide errors of the oligonucleotide pool prior to cloning and passaging through the cells would be useful to include so that the error modes can be better understood.

Nitpicks: Please don't use the phrase 'massive oligos' it is confusing. Please replace with 'large oligonucleotide pools' or something similar.

Reviewer #2 (Remarks to the Author):

This manuscript by Min Hao, et al. reported information storage in live bacterial DNA. The bacterial cells act as the information carrier and amplification media before the extra of the information. The overall concept is interesting, and the data in the manuscript proves the preliminary feasibility of this approach. Understandably, the current study has its limitations and future optimization and improvement are necessary to make this in vivo information storage more robust. For example, there are noticeable information loss and error after multiple rounds of bacterial growth, which could limit the practicality of the amplification via bacterial growth. Nonetheless, despite the limitations, I believe this is a solid manuscript for the journal, and would recommend only a few minor changes.

1) The strategy for random assembly of large DNA pool proposed in this manuscript is interesting. As known, the assembly efficiency highly dependent on the specificity in homology arm sequence, and this proposed random assembly is designed for information storage, in which large sequence complexity will be expected. As observed in experiment of figure 3, the author has defined the assembly as biased process, so there should more discussion about how the sequence complexity contribute to bias in the assembly process and how to deal with it for stable assembly.

2) The DNA oligo material for assembly in this manuscript was synthesized from chip based massive synthesis, however, the molecule copy for each oligo from this high throughput synthesis is not uniform which has been demonstrated by many previous studies, so the unbalanced copy number should influence the random assembly of oligo pool as well. So, the author should explain more how the unbalanced copy number of DNA oligo impact on this assembly and information storage.

3) As presented in figure 3, it is very interesting to find out that the DNA oligo population was stably passaged for multiple generations, but the result also showed that oligo got lost during the passaging, although the DNA losing between the first

and fifth generation is surprisingly small, I think author should give more detail analysis and discussion about how the DNA losing during the passaging.

Referee expertise:

Referee #1: Synthetic DNA

Referee #2: Information storage in DNA

Reviewers' comments:

Reviewer #1 (Remarks to the Author):

Overview: The authors describe a method for using DNA oligonucleotides for information storage in a living bacterial cell. To do this they cloned synthetic oligonucleotides into a vector using various levels of DNA assembly containing 1, 3 or 5 inserts and transformed them into *E. coli*. The authors observed a decrease in oligonucleotide sequence recovery as the cloning complexity increased. This observation is in keeping with anecdotal evidence in DNA synthesis literature. The authors also report that the oligonucleotide libraries remain stable after 5 passages of the culture on media. The authors created two information encoded oligonucleotide libraries containing either 509 or 11520 distinct oligonucleotides which they cloned into a vector and transformed into *E. coli*. The largest library the authors claim is the largest oligonucleotide library ever stored in a living cell.

Comments: Overall the data and conclusions presented by the authors was well organized and clearly stated and supported by the presented materials and figures. The paper could use a good edit to fix some grammatical issues and improve readability. The figures are well done and the captions were highly descriptive and well done.

Response: Thank so much for this kind comments.

While the work described herein does appear to be a logical extension and step-up in complexity from previous in-cell memory storage work we remain skeptical of this and other in vivo methods for large scale storage of information due to the inherent limitations of constructing and transforming DNA into cells. As the authors report, assembling information bearing oligonucleotides into longer constructs which would in effect store more information per vector actually decreased the recoverability of the information from the plasmids isolated. As reported the most efficient method was to clone one oligo at a time. While this inefficiency could be due to the assembly method used and could likely be improved with some optimization the ultimate ceiling for in vivo methods will be the transformation efficiency and the efforts required to recover the information at a given scale compared to in vitro methods.

This being said the author's work does stand as a way point for potential improvements of in vivo DNA information storage and would be of interest to the community.

Response: Thank so much for this kind comment which we totally agree.

To improve this work besides a mild edit for grammar and readability the authors should include an improved discussion of the oligonucleotide encoding scheme and the oligonucleotide information payload design. For example was there sequences that were excluded due to bio-incompatibility issues? Structure? Homopolymers? Cryptic RBSs, etc...

Response: Sorry for this inconvenience caused by writing, we have carefully modified the language through the manuscript with the help of one professional native writer.

We added description for the oligonucleotide information payload design in manuscript **line 133**, and discussion in **line 293**. This is a very important point as reviewer pointed out, the artificially designed DNA sequence may include sequence with biofunction. As revised in manuscript, the scheme of oligo and design strategy were described in detail. One table was added as supplementary figure 3b to list restriction for oligo design including homopolymers, endonuclease recognition sequence and specific sequence. Actually, for minimizing the interference on host cell, the designed oligo has been analyzed to avoid potential bacterial promoter using open bioinformatic tool. Furthermore, for this mixed culture cell strategy, oligo with toxic sequence will not bring damage to others, in comparing with storing all information on one single large fragment. In considering the stable passaging result, it indicated that there is not much biofunction interference generated from the designed sequence.

Also a figure or table showing the oligonucleotide errors of the oligonucleotide pool prior to cloning and passaging through the cells would be useful to include so that the error modes can be better understood.

Response: Thank for this suggestion, we have added the substitution and indel error rate of oligo pool prior to cloning and passaging in revised **figure 3b**, which was obtained by sequencing the master pool only after a few PCR amplification. We also add discussion in **line 197**. All the error rate is the same order of magnitude indicating that the high fidelity of DNA replication in cell is good for DNA storage.

Nitpicks: Please don't use the phrase 'massive oligos' it is confusing. Please replace with 'large oligonucleotide pools' or something similar.

Response: Thanks for this suggestion, we have all the word 'massive oligos' to 'large oligonucleotide pools' in both manuscript (**line 18, 74, 77, ...**) and supplementary materials.

Reviewer #2 (Remarks to the Author):

This manuscript by Min Hao, et al. reported information storage in live bacterial DNA. The bacterial cells act as the information carrier and amplification media before the extra of the information. The overall concept is interesting, and the data in the manuscript proves the preliminary feasibility of this approach. Understandably, the current study has its limitations and future optimization and improvement are necessary to make this in vivo information storage more robust. For example, there are noticeable information loss and error after multiple rounds of bacterial growth, which could limit the practicality of the amplification via bacterial growth. Nonetheless, despite the limitations, I believe this is a solid manuscript for the journal, and would recommend only a few minor changes.

1) The strategy for random assembly of large DNA pool proposed in this manuscript is interesting. As known, the assembly efficiency highly dependent on the specificity in homology arm sequence, and this proposed random assembly is designed for information storage, in which large sequence complexity will be expected. As observed in experiment of figure 3, the author has defined the assembly as biased process, so there should more discussion about how the sequence complexity contribute to bias in the assembly process and how to deal with it for stable assembly.

Response: Thanks for this suggestion. This is a very crucial point, as discussed in the manuscript, the sequence complexity in DNA storage application is much complex than other related bioengineering application. With regrading to the influence of sequence complexity in biased assembly, we have added new discussion in **line 221 and 311** in manuscript. We believed that the biased assembly process could be improved by more effective in vitro DNA homology assembly reagent and DNA origami assembly process could also be helpful to improve the large number oligo assembly.

2) The DNA oligo material for assembly in this manuscript was synthesized from chip based massive synthesis, however, the molecule copy for each oligo from this high throughput synthesis is not uniform which has been demonstrated by many previous studies, so the unbalanced copy number should influence the random assembly of

oligo pool as well. So, the author should explain more how the unbalanced copy number of DNA oligo impact on this assembly and information storage.

Response: This is a very crucial point. As discussed in the manuscript, due to the scale of DNA oligo usage, DNA information storage highly dependent on the chip based large number DNA oligo synthesis. However, the copy number is not uniform as discussed in the introduction of this manuscript. In revised manuscript, we added more discussion in **line 341** for the influence of unbalanced copy number in the large nucleotide oligo pool. We believed that more technology could be develop to modify the unbalanced copy distribution. Actually, we are working on this crucial issue in our ongoing project.

3) As presented in figure 3, it is very interesting to find out that the DNA oligo population was stably passaged for multiple generations, but the result also showed that oligo got lost during the passaging, although the DNA losing between the first and fifth generation is surprisingly small, I think author should give more detail analysis and discussion about how the DNA losing during the passaging.

Response: thanks for this constructive comment. As discussed in manuscript, the dropout during the successive passaging is very small. Due to the mixed culture strategy, only small piece was transformed to each cell, the growth burden from these exogenous DNA could be small enough to be ignored. That is the huge advantage of this mixed culture strategy in comparing large fragment DNA. Following reviewer's suggestion, we have added more discussion in **line 342** in revised manuscript. The oligo dropout could be considered as a random and occasional thing, by which oligo with small copy number is prone to dropout during the assembly and culture process. Associated with the second comment, we believe that the adjusted oligo pool with more even copy number could largely decrease this dropout. We hope we can go back lab soon and push related project moving forward.

REVIEWERS' COMMENTS:

Reviewer #1 (Remarks to the Author):

The Authors have addressed the weaknesses/comments from the previous review and we believe that the manuscript is better for it. We recommend this manuscript for publication.

Reviewer #2 (Remarks to the Author):

Comments and concerns had be satisfactorily addressed by the author's revision. Recommend for acceptance.

Reviewer #1 (Remarks to the Author):

The Authors have addressed the weaknesses/comments from the previous review and we believe that the manuscript is better for it. We recommend this manuscript for publication.

Responses: thanks your effort reviewing our manuscript

Reviewer #2 (Remarks to the Author):

Comments and concerns had be satisfactorily addressed by the author's revision. Recommend for acceptance.

Responses: thanks your effort reviewing our manuscript

REVIEWERS' COMMENTS:

Reviewer #1 (Remarks to the Author):

The Authors have addressed the weaknesses/comments from the previous review and we believe that the manuscript is better for it. We recommend this manuscript for publication.

Reviewer #2 (Remarks to the Author):

Comments and concerns had be satisfactorily addressed by the author's revision. Recommend for acceptance.